# Plant Monoterpenes and Essential Oils as Potential Anti-Ageing Agents: Insights from Preclinical Data

**DOI:** 10.3390/biomedicines12020365

**Published:** 2024-02-04

**Authors:** Mónica Zuzarte, Cátia Sousa, Jorge Alves-Silva, Lígia Salgueiro

**Affiliations:** 1Univ Coimbra, Faculty of Pharmacy, Azinhaga de S. Comba, 3000-548 Coimbra, Portugal; jmasilva@ff.uc.pt (J.A.-S.); ligia@ff.uc.pt (L.S.); 2Univ Coimbra, Coimbra Institute for Clinical and Biomedical Research (iCBR), Faculty of Medicine, Azinhaga de S. Comba, 3000-548 Coimbra, Portugal; 3Clinical Academic Centre of Coimbra (CACC), 3000-548 Coimbra, Portugal; 4iNOVA4HEALTH, NOVA Medical School, Faculdade de Ciências Médicas (NMS/FCM), Universidade Nova de Lisboa, 1159-056 Lisboa, Portugal; catia.moreirasousa@nms.unl.pt; 5Centro Clínico e Académico de Lisboa, 1156-056 Lisboa, Portugal; 6Univ Coimbra, Chemical Engineering and Renewable Resources for Sustainability (CERES), Department of Chemical Engineering, 3030-790 Coimbra, Portugal

**Keywords:** autophagy, dysbiosis, genomic instability, inflammation, mitochondrial dysfunction, senescence

## Abstract

Ageing is a natural process characterized by a time-dependent decline of physiological integrity that compromises functionality and inevitably leads to death. This decline is also quite relevant in major human pathologies, being a primary risk factor in neurodegenerative diseases, metabolic disorders, cardiovascular diseases and musculoskeletal disorders. Bearing this in mind, it is not surprising that research aiming at improving human health during this process has burst in the last decades. Importantly, major hallmarks of the ageing process and phenotype have been identified, this knowledge being quite relevant for future studies towards the identification of putative pharmaceutical targets, enabling the development of preventive/therapeutic strategies to improve health and longevity. In this context, aromatic plants have emerged as a source of potential bioactive volatile molecules, mainly monoterpenes, with many studies referring to their anti-ageing potential. Nevertheless, an integrated review on the current knowledge is lacking, with several research approaches studying isolated ageing hallmarks or referring to an overall anti-ageing effect, without depicting possible mechanisms of action. Herein, we aim to provide an updated systematization of the bioactive potential of volatile monoterpenes on recently proposed ageing hallmarks, and highlight the main mechanisms of action already identified, as well as possible chemical entity–activity relations. By gathering and categorizing the available scattered information, we also aim to identify important research gaps that could help pave the way for future research in the field.

## 1. Introduction

The number and proportion of people aged 60 years or older is increasing at an unprecedented rate and, by 2050, will reach up to 2.1 billion, as stated by WHO [1]. Indeed, population ageing is the most global demographic trend, conveying several public challenges primarily in the health sector. The ageing process is associated with an impairment of several body functions and an increase in age-related disorders, including neurodegenerative, cardiovascular, cancer and metabolic diseases [2]. Several factors including genetics, lifestyle and environmental features are key in this inevitable gradual process, and interventions that promote health and longevity, thus increasing lifespan, are a matter of current interest and research.

In the last years, several strategies have been proposed to delay ageing, namely caloric restriction, nutritional interventions and microbiota transplantation [3,4]. In addition, specific treatments including depletion of senescent cells, stem cell therapy, antioxidant and anti-inflammatory treatments, and hormone replacement therapy have been used in order to promote healthy ageing and longevity [5]. The identification of potential drugs for age-related diseases such as metformin, rapamycin, resveratrol and senolytics is also a hot topic in the field, although several recognized limitations and side effects raise some concerns that need further clarification [2].

The main cellular and molecular mechanisms involved in biological ageing enabled the identification, in 2013, of ageing hallmarks [6]. Later, in 2022, a research symposium held in Copenhagen considered the addition of new hallmarks to the existing list, bearing in mind their role in the ageing process [7]. The accepted current list now includes 12 ageing hallmarks that comprise genomic instability, telomere attrition, epigenetic alterations, loss of proteostasis, compromised autophagy, deregulated nutrient sensing, mitochondrial dysfunction, cellular senescence, stem cell exhaustion, altered intercellular communication, chronic inflammation and dysbiosis [8]. These hallmarks have proven to be very useful in the identification of intervention targets, primarily in age-related diseases with promising results in preclinical models [9,10] and validations in clinical trials [11]. Among the tested compounds, natural compounds have attracted researchers’ interest, with recent studies pointing out their effect on critical ageing pathways such as mammalian target of rapamycin (mTOR), adenosine monophosphate-activated kinase (AMPK), Sirtuin 1 (SIRT1) and tumor protein p53, as reviewed elsewhere [12]. Nevertheless, the majority of the studies resort to in vitro approaches, lacking in-depth analysis of individual compounds and specific effects on these pathways [12]. Bearing this in mind, we sought to review the potential of volatile monoterpenes, a group of compounds highly prevalent in aromatic plants. These compounds are valued by several industries due to their bioactive potential, namely antimicrobial, antioxidant, anti-inflammatory, analgesic, antinociceptive and anticancer effects, as reviewed elsewhere [13]. Importantly, many monoterpenes are considered safe for consumer use and have a generally recognized as safe (GRAS) status by the Food and Drug Administration (FDA), being included in many foods and beverages.

In the present review, an up-to-date overview of the effect of monoterpenes is provided, considering the current hallmarks of ageing. In addition, whenever known, the main mechanisms of action underlying the observed effects are stated, and a possible chemical entity–bioactivity relation is attempted. Importantly, the main gaps in the current knowledge are pointed out to help guide future research in the field.

## 2. Aromatic Plants and Volatile Compounds

The term aromatic plants refers to plants that produce volatile compounds. These compounds can be extracted from the plant source as a volatile mixture called essential oil. To comply with regulatory requirements, in particular those from the International Standard Organization on Essential Oils [14] and the European Pharmacopoeia [15], essential oils are obtained from the plant raw material by hydrodistillation, steam distillation or dry distillation, or by a suitable mechanical process in the case of *Citrus* fruits. Several monographs on essential oils are available, for example, in the European Pharmacopoeia [15] or the European Medicine Agency [16], which highlights their importance in the pharmaceutical industry. The main types of compounds present in essential oils are monoterpenes and sesquiterpenes. Monoterpenes are isoprene dimers with carbon and hydrogen atoms (C_10_H_16_), such as α-pinene, limonene, camphene, myrcene and *p*-cymene. Monoterpenoids include in their chemical structure functional groups like alcohols (e.g., linalool, geraniol, menthol), esters (e.g., linalyl acetate, geranyl acetate, neryl acetate), ketones (e.g., camphor, menthone, carvone), aldehydes (e.g., geranial, neral), ethers (e.g., 1,8-cineole) or phenols (e.g., carvacrol, thymol) [17]. Monoterpenes are synthetized from geranyl pyrophosphate by monoterpene synthases and, according to their structure, can be classified as acyclic, monocyclic or bicyclic. These volatile compounds generally present a pleasant aroma and can be found in fruits, vegetables, spices and herbs, as well as in different plant parts like leaves, flowers, seeds and roots. They are appreciated as aromatic and flavoring agents in cosmetic, pharmaceutical and food industries. Also, emerging evidence shows that monoterpenes present several biological properties including antimicrobial, antioxidant, anti-inflammatory and cardioprotective, as previously reviewed [18]. These findings have led to the incorporation of monoterpenes in distinct products, such as diet supplementation, animal feed additives, food packaging and biopesticides, among others. Interestingly, modifications of these compounds have also been considered with growing evidence on the biological and medical application of monoterpene derivatives [19].

## 3. Anti-Ageing Potential of Low-Molecular-Weight Terpenes

### 3.1. Ageing Hallmarks

Ageing is characterized by a progressive loss of physiological integrity and impaired function that increases vulnerability to pathology or age-associated diseases, such as neurodegenerative diseases, metabolic disorders, cardiovascular diseases and musculoskeletal disorders [6]. Ageing per se does not cause these diseases but favours their clinical manifestation [20], since the mechanisms driving ageing and those driving age-associated diseases largely overlap [21].

In 2013, López-Otín and colleagues proposed nine hallmarks of ageing, namely genomic instability, telomere attrition, epigenetic alterations, loss of proteostasis, deregulated nutrient sensing, mitochondrial dysfunction, cellular senescence, stem cell exhaustion and altered intercellular communication. Accordingly, to be considered a hallmark, these processes should manifest during ageing; their aggravation, resorting to experimental models, should accelerate the ageing process, and experimental improvement should delay the ageing process [6]. During the last years, extensive research in the ageing field and its hallmarks has been carried out, leading to an update of the initially proposed list [8]. Thus, three new hallmarks were added, namely disabled macroautophagy, chronic inflammation and dysbiosis. Indeed, López-Otín and colleagues considered that macroautophagy does not only affect proteins but also targets organelles and non-proteinaceous macromolecules, thereby being separated from the hallmark loss of proteostasis. Moreover, the authors also suggested that altered cellular communication was a vast hallmark, so chronic inflammation and dysbiosis were separated from it [8].

Independently of the number of hallmarks considered, there is an established hierarchy among them [6,8]. Thus, the hallmarks are categorized as: primary, which reflects the damage, inflicted to the genome, telomeres, epigenome, proteome and organelles; antagonistic, which reflects the response to the damage and includes deregulated nutrient sensing, mitochondrial dysfunction and cellular senescence; and integrative, which considers the end result, leading to stem cell exhaustion, intercellular communication alterations, chronic inflammation and dysbiosis. Consequently, the integrative hallmarks are responsible for the functional decline associated with ageing [6,8].

Interestingly, the hallmarks of ageing are not independent, implying that experimental accentuation or attenuation of one specific hallmark will consequently affect the others [8]. This is an interesting aspect to bear in mind, regarding the discovery and/or development of new drugs to tackle ageing and related diseases.

### 3.2. Monoterpenes and Essential Oils in Ageing Hallmarks

In this section, a compilation of the main studies reporting the effect of volatile monoterpenes on ageing hallmarks is presented. The search was carried out in the PubMed database combining the words “monoterpene” and each hallmark of ageing, without date restrictions. The results obtained are presented in the following subsections. However, for inflammation, due to the high number of studies on this topic, a more targeted search was carried out, as detailed in Section 3.2.3. For each hierarchized hallmark, a brief overview is provided followed by a table summarizing the studies organized in alphabetic order of the compound name. Information on the in vitro or in vivo study model used and the main effects reported is shown. In addition, evidence regarding the effect of essential oils on these hallmarks is also provided.

#### 3.2.1. Primary Hallmarks

##### Genomic Instability

Genomic instability is generally characterized by genetic lesions, which include point mutations, deletions, translocations, telomere shortening, single and double strand breaks, chromosomal rearrangements, defects in nuclear architecture and gene disruption. These lesions can be caused by extrinsic (e.g., chemical, physical and biological agents) and intrinsic (e.g., DNA replication errors, chromosome segregation defects, oxidative processes and spontaneous hydrolytic reactions) sources. To maintain the genomic integrity and stability, organisms have a complex network of DNA repair mechanisms. Nevertheless, excessive DNA damage or insufficient repair favours the ageing process [6,8]. Different tools have been developed to detect and analyse genomic instability, particularly in cancer models, and include DNA damage, micronuclei development, chromatin and anaphase bridge formation, as well as aneuploidy [22]. Studies reporting the in vitro effect of monoterpenes on genomic instability are summarized in Table 1.

Although the primary hallmarks of ageing are the primary causes of cell damage, only two in vitro studies reported the negative effects of monoterpenes on genomic instability, namely α-pinene and D-limonene. These monoterpenes are quite frequent in several essential oils, α-pinene being characteristic of *Pinus* spp. (pine) essential oils and D-limonene highly prevalent in *Citrus* spp. essential oils. The results presented in Table 1 suggest that α-pinene and D-limonene favour genomic instability. Indeed, α-pinene induces genomic instability by promoting mitotic alterations and reactive oxygen species (ROS) production, which causes DNA damage [23]. On the other hand, D-limonene affects the cell division by preventing the assembly of mitotic spindle microtubules, which then affect chromosome segregation and cytokinesis, consequently leading to aneuploidy [24].

Regarding other primary hallmarks, a study on the effect of *Thymus vulgaris* (thyme) essential oil on blood telomere attrition, in chronologically aged C57BL/6J mice, was carried out [25]. Interestingly, the authors reported a beneficial effect and demonstrated that mice supplemented with this essential oil (0.2% *w*/*w*) showed higher survival rates and significantly longer blood telomere lengths than the control mice. Although the study did not assess isolated compounds, the authors suggest that the phenolic monoterpene thymol and the monoterpene hydrocarbon *p*-cymene, present in high amounts in the oil, may primarily contribute to the observed telomere-protecting effect, due to their antioxidant and anti-inflammatory potentials [25]. Thus, it seems that the aromatic ring might be responsible for the different outcomes observed in these primary hallmarks, particularly genomic instability and telomere attrition. However, further studies are needed to verify this hypothesis.

#### 3.2.2. Antagonistic Hallmarks

##### Mitochondrial Dysfunction

Mitochondria are organelles that play a vital role in homeostasis maintenance [26]. Deterioration in mitochondria function increases with ageing due to the occurrence of diverse mechanisms, namely accumulation of mitochondrial DNA mutations and changes in their dynamics. These alterations might compromise the mitochondria contribution to cellular bioenergetics, increasing the production of ROS and, consequently, promoting the mitochondria membranes permeabilisation. The latter phenomenon leads to inflammation and cell death [8].

Mitochondrial dysfunction can be assessed both in vitro and in vivo. For example, isolated mitochondria can be used to assess mitochondria respiratory capacity, defined as the increase in respiration rate in response to ADP. In cells, studies include the measurement of several parameters such as ATP production rate, proton leak rate, coupling efficiency, maximum respiratory rate, respiratory capacity ratio and spare respiratory capacity [27]. Table 2 compiles both in vitro and in vivo studies on the effect of monoterpenes on mitochondrial dysfunction.

Combining both in vitro and in vivo studies, 1,8-cineole (synonym of eucalyptol), carvacrol, geraniol and perillyl alcohol are the most studied compounds. These compounds can be found in the essential oils of several aromatic plants; for example, 1,8-cineole is very common in *Eucalyptus globulus*, while carvacrol is characteristic of *Origanum* species, geraniol is abundant in *Rosa* spp. and perillyl alcohol occurs in different *Mentha* species. A closer analysis of the information presented in Table 2 shows that the studied monoterpenes have a beneficial effect, thus inhibiting mitochondrial dysfunction in both in vitro and in vivo models. The majority of the tested compounds present in their structure a hydroxyl group or a phenol group that might contribute to the observed effects. In general, the tested monoterpenes decreased ROS production, thus showing a beneficial effect, by restoring the mitochondrial membrane potential as shown for 1,8-cineole [29,31] and geraniol [34]. The increase in mitochondrial membrane potential is also beneficial as ATP synthase activity is enhanced [51]. Indeed, most studies reporting an increase/restoration of mitochondrial membrane potential also demonstrate an increase in ATP production, such as geraniol [34] and carvacrol [33]. The activity of the mitochondrial respiratory chain complexes can also contribute to the maintenance of the mitochondrial membrane potential through proton flux [51]. These effects were reported, for example, for geraniol [34] and linalool [37].

Besides monoterpenes, several studies have reported the effects of essential oils on mitochondrial dysfunction, although in distinct contexts not directly related to ageing. Indeed, the majority of the studies are carried out in cancer models, thus showing an overall beneficial effect but through an enhancement of mitochondrial dysfunction. For example, *Schisandrae semen* essential oil induced ROS- and caspase-dependent cell death involving mitochondrial dysfunction and nuclear translocation of mitochondrial pro-apoptosis proteins in human leukaemia U937 cells [52]. The essential oil of *Alpinia officinarum* promoted lung cancer regression by triggering mitochondrial membrane potential dysfunction and activating caspase-3 cleavage, inducing cell apoptosis [53]. *Cinnamomum cassia* essential oil showed anti-oral cancer activity in HSC-3 cells, through a significant increase in ROS production [54]. The cytotoxic potential of *Croton heliotropiifolius* on different human cancer cell lines, namely leukaemia, colon, melanoma and glioblastoma, was also related to oxidative stress, culminating in mitochondrial respiratory dysfunction and DNA damage, triggering cell death via apoptosis [55]. In the context of cancer, some monoterpenes like carvacrol [56] and hinokitiol (synonym of β-thujaplicin) [57] are also able to potentiate mitochondrial dysfunction.

In addition to essential oils’ anticancer effects, their antimicrobial properties have also been related to mitochondrial dysfunction, being the mitochondrial impairment referred to as the mechanism of action that compromises infection. These effects were shown, for example, in *Rosmarinus officinalis* essential oil against *Candida albicans* [58], *Piper nigrum* essential oil against *Aspergillus flavus* [59], and *Anethum graveolens* essential oil against both *A. flavus* [60] and *C. albicans* [61]. Moreover, insecticidal properties of essential oils through mitochondrial impairment in insects have been reported, for example, for *Eugenia uniflora* essential oil in *Drosophila melanogaster* [62] and *Melaleuca alternifolia*, that prevented bioenergetics dysfunction in the spleen of silver catfish naturally infected with *Ichthyophthirius multifiliis* [63].

##### Cellular Senescence

Cellular senescence is a state of permanent cell cycle arrest occurring when telomere length decreases below a critical size, in response to different damaging stimuli, such as oxidative stress and DNA damage [64,65]. This is an important barrier mechanism to tumorigenesis as it limits the growth of potentially oncogenic cells [65]. Moreover, cellular senescence plays a key role in some physiological processes such as embryogenesis, tissue remodelling and repair [65]. The senescent phenotype is characterized by chronic activation of the DNA damage response, upregulation of Cyclin-Dependent Kinase (CDK) inhibitors (e.g., p16INK4a, p15INK4b and p21CIP), apoptosis resistance, altered metabolic rates, endoplasmic reticulum stress and increasing secretion of pro-inflammatory and tissue remodelling factors, known as the Senescence-Associated Secretory Phenotype (SASP) or senescence-messaging secretome (SMS) [64]. The SASP is quite important to ensure the efficient growth arrest by autocrine signalling, in particular immediately after senescence induction, to signal senescent cells for clearance by the immune system and, consequently, for tissue repair and remodelling [66,67]. Moreover, the SASP is highly heterogeneous [64] and can include a wide range of cytokines, chemokines, proteases, growth factors [66] and non-macromolecular elements [65]. Although various signalling pathways are involved in the regulation of SASP components, most of them converge to NF-κB activation [65,66]. Morphologically, senescent cells show structural aberrations, including an enlarged and more flattened shape; modified composition of the plasma membrane due to an upregulation of caveolin-1; increased lysosomal content which is directly correlated with higher senescence-associated β-galactosidase activity (pH 6.0); accumulation of mitochondria and nuclear changes like loss of Lamin B1 and formation of senescence-associated heterochromatin *foci* [64]. Excessive and anomalous accumulation of senescent cells in tissues has a negative impact on homeostasis mainly via SASP [67]. This phenomenon can be detrimental to regenerative capacities and generate a disruptive pro-inflammatory environment that is favourable for the onset and progression of a variety of age-related diseases [64,67]. The accumulation of these cells can occur during ageing [65].

González-Gualda and colleagues proposed that, at least, three different types of markers that confirm (i) cell cycle arrest, (ii) increased lysosomal compartment or another global senescent-related structural change and (iii) an additional trait specific for the particular type of senescence being assessed (such as SASP or DNA damage features) should be considered to assess senescence [68]. Monoterpenes known to possess anti-senescent effects are summarized in Table 3.

Regarding the effect of monoterpenes on cellular senescence, only five compounds have been tested for their senolytic or senomorphic effects. Senolytic compounds are those that are able to effectively kill senescent cells, by modulating several associated pathways, such as p21/p53 and by inducing apoptosis, whereas senomorphic compounds are responsible for inhibiting SASP, through modulation of associated pathways, such as NF-κB, mTOR and JAK/STAT [73]. For instance, α-pinene can be considered a senolytic compound due to its effect on the p21/p53 axis [71], thus increasing DNA repair, preventing the entrance in cell cycle arrest and, consequently, senescence [74]. D-Limonene also acts as a senolytic, probably by inducing cell death by apoptosis as it prevents the nuclear translocation of p53 [35], which is known to induce mitochondria-dependent apoptosis [75]. Myrcene, on the other hand, showed a senomorphic effect through a direct decrease of IL-6 secretion [39]. Secretion of matrix-degrading mediators, such as metalloproteinases, is also characteristic of SASP [73]. Having this in mind, α-pinene [72], camphor [69], myrcene [39] and hinokitiol [70] are all considered senomorphics due to their inhibitory effect in the synthesis or secretion of metalloproteinases. The compounds, referred to in Table 3, are present in several essential oils. Camphor is typically encountered in *Cinnamomum camphora*, hinokitiol is present in *Chamaecyparis* spp., D-limonene is characteristic of *Citrus* fruit peels, myrcene occurs in *Cannabis sativa* and in several other essential oils and α-pinene is one of the main compounds in *Pinus* spp.

Interestingly, other studies assessed the effect of monoterpenes on features related to senescence but in a cancer context, similar to that reported in the previous section. In these cases, an enhancement in cellular senescence is the beneficial effect as it negatively affects the cancer cell growth. For example, a nanoemulsion of carvacrol induced cell senescence leading to cell cycle arrest by reducing CDK2, CDK4, CDK6, Cyclin E, Cyclin D1 and enhancing p21 protein expressions. Also, an increase in SA-β-galactosidase activity was observed [56]. Hinokitiol decreased cell proliferation and increased DNA damage and phase S cell cycle arrest in osteosarcoma cells U2OS and MG63 [76], while increasing γH2AX-positive cells in a xenograft mice model [77]. Other monoterpenes assessed in the context of cancer include 1,8-cineole that was effective in increasing phase G0/G1 cell cycle arrest, while decreasing CDK2, CDK4 and CDK6 and cyclin A protein levels. This compound also increased p27 and cyclin D1 protein levels and SA-β-galactosidase activity [78]. Geraniol increased phase sub G1 cell cycle arrest in prostate cancer cells PC3 [79]. Menthol and α-phellandrene increased p53 levels in the leukaemia cell lines NB4 and Molt 4 [80] and nuclei levels of p53 while decreasing its cytoplasmic levels in the hepatocellular cell line J5, respectively [81].

Regarding essential oils, several studies have assessed their effect on cellular senescence. For example, *Thymbra capitata* decreased SA-β-galactosidase activity in fibroblasts treated with the senescence inducer etoposide [82]. Similar results were reported for *Santolina rosmarinifolia* [83], *Salvia aurea* [84] and *Ferulago lutea* [85]. For the latter, additional senescence features (p21/p53 protein levels and the nuclear accumulation of γH2AX) were also decreased following essential oil treatment [85]. In addition, *Eucalyptus globulus* essential oil showed anti-senescent effects in both etoposide-stimulated keratinocytes (HaCaT) and fibroblasts (NIH/3T3). Thus, in the presence of the oil, the number of senescent cells decreased as well as the levels of p53 [86]. Other essential oils showed beneficial properties *in vitro*, senescence being induced by distinct agents. For example, several *Cistus* spp. oils significantly attenuated UVB-induced cellular senescence in a keratinocyte cell line (HaCaT) [87], while *Cymbopogon nardus* and *Cymbopogon citratus* oils showed cytoprotective properties on doxorubicin-induced senescence in the fibroblast-like kidney cell line (Vero) and fibroblasts (NIH/3T3) [88].

An in vivo study was also carried out in diabetic mice with *Alpiniae zerumbet* administered orally, ameliorating vascular endothelial cell senescence by activating PPAR-γ signalling [89].

#### 3.2.3. Integrative Hallmarks

##### Disabled Macroautophagy

Autophagy is a physiological cellular mechanism whereby cytoplasmic material is delivered to the lysosome for degradation [90]. Macroauthophagy is the most well characterized and involves the sequestration of cytoplasmic material, including soluble macromolecules and organelles, into a double- or multi-membrane-bound structure, named autophagosome. These structures will then fuse with the lysosome, promoting degradation and recycling of its content [90]. This section will focus only on macroautophagy, henceforth referred to as autophagy. Under normal conditions, constitutive autophagy occurs as a housekeeping mechanism [91]. However, autophagy can be activated during starvation and stress conditions such as hypoxia, increased ROS production, DNA damage, protein aggregates, damaged organelles or intracellular pathogens, in order to restore homeostasis [92]. Moreover, it is well established that autophagy declines with age [6,93].

Several approaches have been developed to assess autophagy, namely immunoblotting of canonical markers like LC3 and p62, detection of autophagosomes by fluorescence microscopy or autophagosome maturation resorting to mRFP-GFP fluorescence microscopy, although a combined use of several methods is recommended [94]. Table 4 includes studies on the effect of monoterpenes on autophagy.

In what concerns the effect of monoterpenes on autophagy, both in vitro and in vivo studies are available and, similarly to that shown in the mitochondrial dysfunction section, the majority of the tested compounds present in their structure a hydroxyl group or a phenol group that might contribute to the observed effects. For this hallmark, several compounds have been tested in vivo: for example, borneol that occurs in *Salvia rosmarinus*; carvacrol present in *Origanum* spp.; 1,8-cineole characteristic of *Eucalyptus* spp.; citronellol found in *Cymbopogon nardus*; geraniol abundant in several flowers, such as *Rosa* spp.; myrcene present, for example, in *Cannabis sativa*; and thymol frequent in *Thymus* spp.

Regarding their effect on autophagy, monoterpenes can be divided into those that promote autophagy and those that decrease it. Although these effects seem contradictory, it is important to refer that the disease model used differs between the studies, and therefore, the monoterpenes increase or decrease autophagy in order to restore homeostasis. For instance, carvacrol [43,95], 1,8-cineole [29,31], hinokitiol [98], menthol [100] and borneol [102] induced an increase in autophagy, by enhancing the ratio LC3B-II/LC3B-I, LC3 and/or Beclin-1 levels, whereas decreasing that of p62. On the other hand, some studies showed the opposite effect. For example, 1,8-cineole [96,97], hinokitiol [99], thymol [101], citronellol [103], geraniol [104] and myrcene [105] decreased the ratio of LC3B-II/I, LC3 and/or Beclin-1 levels, while increasing that of p62 and/or p-mTOR levels, consequently reducing autophagy.

Besides the studies highlighted in Table 4, the effect of monoterpenes on autophagy has also been reported in cancer cells. The majority of the compounds tested in this setting increase autophagy, thus suppressing tumorigenesis by inhibiting cell survival and inducing cell death. For example, hinokitiol was largely assessed and increased autophagy in vitro in hepatoblastoma cell line HepG2 [57], HeLa cervical cancer cells [107], U2OS and MG63 osteosarcoma cells [76] and H1975 lung cancer cells [77], through an increase in LC3 levels, Beclin-1 levels and/or autophagosome formation, among other autophagy markers. In addition, α-phellandrene induced autophagy in human hepatocellular carcinoma (J5) cells by regulating mTOR and LC3-II expression, as well as p53 signalling. Indeed, an increase in autophagic vacuoles formation was observed together with a decrease in PI3K, Akt and mTOR protein levels and an increase in Beclin-1 and LC3-II [81]. In addition, α-thujone induced oxidative stress and autophagy in a glioblastoma cell line (T98G), as shown by an increase in autophagolysosome formation, an increase in LC3-II/LC3-I ratio, as well as an increase in the levels of LC3-II, Beclin-1, Atg3, Atg5 and Atg7 [108]. Furthermore, terpinen-4-ol induced an accumulation of LC3-I/II, Atg5 and Beclin-1, as well as regulatory proteins required for autophagy in human leukemic HL-60 cells [109]. In vivo studies also pointed out the effects of hinokitiol on the inhibition of xenograft tumour growth in association with DNA damage and autophagy, as shown by the expression of γ-H2AX and LC3 in the tumour tissue [77].

Essential oils also affect autophagy, and have been assessed in distinct contexts, namely cancer and cell toxicity, among others. Concerning anticancer effects, for example, the essential oil of *Origanum montana* inhibited colony growth of human HT-29 colorectal cancer cells by inducing protective autophagy, associated with downregulation of the mTOR/p70S6K pathway and apoptotic cells death, via the activation of the p38 MAPK signalling pathway [110]. In what concerns cell toxicity, the essential oil of *Schisandra chinensis* was able to attenuate acetaminophen-induced liver injury by alleviating oxidative stress and activating autophagy, showing an upregulation of hepatic LC3-II and a decrease in p62 in mice with an overdose of acetaminophen [111]. Moreover, the essential oil of *Acorus tatarinowii* ameliorated Aβ-induced toxicity in *Caenorhabditis elegans* by maintaining protein homeostasis through the autophagy pathway regulated partly by hsf-1 and sir-2.1 genes [112].

##### Inflammation

According to Antonelli and Kushner (2017), inflammation is defined as “an innate immune response to harmful stimuli such as pathogens, injury and metabolic stress that aims to restore homeostasis” [113]. In basal conditions, tissues are maintained in the homeostatic state with the aid of tissue-resident macrophages, when it is required. Nevertheless, in noxious conditions, tissues undergo stress and can malfunction. In the case of considerable changes, adaptation to these conditions requires the help of tissue-resident macrophages or the recruitment of other macrophages and may require small-scale delivery of additional leukocytes and plasma proteins. This response has characteristics that are intermediate between basal and inflammatory states and was termed *para-inflammation* [114]. *Para-inflammation* is not a classic form of inflammation triggered by exogenous tissue injury or infection, but it is switched on by tissue malfunction in order to promote its adaptation to a harmful environment and to maintain its adequate functionality. This state is characterized by a low-grade/subclinical immune reaction [114]. Nevertheless, if the tissue malfunction is present for a sustained period, *para-inflammation* progresses into a chronic low-grade inflammation (pathophysiological *para-inflammation*) [115,116]. Ageing and related diseases are characterized by the presence of chronic low-grade inflammation [8,114]. Moreover, chronic inflammation might be a consequence of alterations in other hallmarks of ageing and vice-versa, representing a vicious cycle [8,117]. This hallmark is, by far, the most addressed with a PubMed bibliographic search on “monoterpenes” and “inflammation” encountering more than 1500 studies. Therefore, only the studies performed in the last 4 years that assessed the mechanism of action associated to the observed effect were considered, being summarized in Table 5. The anti-inflammatory potential of monoterpenes is usually studied by evaluating the expression of pro-inflammatory mediators (e.g., IL-1β, IL-6, TNF-α, iNOS, NO and COX-2), anti-inflammatory mediators (e.g., IL-10) and the impact on signalling pathways relevant to the inflammation process, particularly NF-κB, MAPKs and Nrf2.

As shown in Table 5, monoterpenes with different chemical structures are able to impact inflammation. Previously, a standardized analysis of the anti-inflammatory activity of *p*-menthane-derived monoterpenes, namely (R)-(+)-limonene, (S)-(−)-limonene, (S)-(+)-carvone and (R)-(−)-carvone, was carried out by Sousa and colleagues. Considering the carbon numbering relative to the common precursor (limonene), the authors highlighted that the presence of an oxygenated group at C6 conjugated to a double bond at C1 and an isopropenyl group and S configuration at C4 are the major chemical features relevant for activity and potency [164]. Nevertheless, further studies are needed to clarify the structure–activity relation of these molecules using similar models and endpoints.

Most of the studies herein reported assess the effect of monoterpenes on the NF-κB signalling pathway as this transcription factor is the master regulator of inflammation [117]. For instance, carvacrol [118], citral [122], geraniol [126], perillyl alcohol [129,130], thymol [132], among others (Table 5) interfered with the canonical activation of NF-κB, blocking its nuclear translocation and, consequently, its transcriptional activity. On the other hand, some monoterpenes did not interfere directly with NF-κB activation but promoted the activation of other molecules, namely SIRT1 and Nrf2, which antagonize the transcriptional activity of NF-κB [119,120,121,131]. Examples of these compounds include (S)-(+)-carvone [119], (R)-(−)-carvone [120], 1,8-cineole [121] and perillaldehyde [131]. Other signaling pathways were also evaluated, namely MAPK and inflammasome NLRP3 that include menthol [128] and hinokitiol [127], respectively.

Aside from the studies presented in Table 5, many others were found in the bibliographic search reporting the anti-inflammatory activity of monoterpenes by evaluating the expression of pro-inflammatory and/or anti-inflammatory mediators, as these are the most common endpoints found in the inflammatory response. Examples include carvacrol [165,166,167,168], 1,8-cineole [169,170], citral [171,172], isopulegol [173], linalool [174,175], linalyl acetate [174] and thymol [176,177,178].

Similar to monoterpenes, many studies also report the anti-inflammatory potential of essential oils. Indeed, the combined search of “essential oil” and “chronic inflammation” brings up 37 reviews in PubMed. Regarding original studies, the essential oil of *Thymus vulgaris* was assessed in ageing-induced brain inflammation in chronologically aged C57BL/6J mice and showed a significantly lower gene expression of the pro-inflammatory cytokine Il6 in the hippocampus and lower Il1b expression in the liver and cerebellum, [25]. Many other studies have been performed not directly related to ageing, with some genera standing out as the most studied. These include essential oils from *Cinnamomum* spp. [179,180], *Lavandula* spp. [181,182], *Thymus* spp. [183,184] and *Citrus* spp. [185,186,187].

##### Dysbiosis

Microbiota includes the microorganisms (e.g., bacteria, fungi, viruses and parasites) that inhabit all parts of our body and can affect human health. The composition and tissue colonization by microbiota are affected by external and internal factors. The former include mainly infant-related factors, such as the mode of delivery, the gestation time and the feeding type, as well as the type of diet and use of antibiotics. The latter includes host internal factors, particularly genetics and the immune system [188]. Importantly, alterations in the microbiome have been recently associated with the emergence of several chronic diseases such as cancer, diabetes, cardiovascular diseases, as well as some psychological disorders, for example, schizophrenia [189]. Moreover, ageing also affects microbiota, particularly gut microbiota, and is associated with microbiome disturbance, named dysbiosis, which is characterized by a shift in microbiota populations and the loss of diversity [8,190]. Indeed, ageing-induced dysbiosis has been associated with the development of several diseases such as lung diseases [191], cardiovascular diseases [192], glaucoma [193] and neurological disorders [194].

Microbiome can be assessed resorting to a plethora of technologies, particularly gene marker analysis, shotgun metagenomics, metabolomics, metaproteomics and metatranscriptomics [195]. Although gene marker analysis that resorts to a targeted sequencing method encompassing the 16S ribosomal RNA gene for bacteria and internal transcribed spacer (ITS) region for fungi is the most common technique, it fails to include other microorganisms, such as virus. Having that in mind, shotgun metagenomics was developed, which resorts to untargeted sequencing methods that allow the capture of the full repertoire of genetic information from a sample, detecting the presence of several microbes. The remaining technologies rely on the detection of expressed microbiome-related genes in the sample (metatranscriptomics), on the small metabolites and their interaction with the host and vice versa (metabolomics) and on the proteins present in the sample (metaproteomics) [195]. The keyword combination of “dysbiosis” and “monoterpenes” only gave back one result, as shown in Table 6. The authors of the study resorted to gene marker analysis by sequencing the 16S ribosomal gene in order to identify the main bacteria found in the mice belonging to the *Bacteriodetes* and *Firmicutes* divisions.

The study herein reported shows that geraniol is able to prevent colitis-associated dysbiosis in the dextran sulphate sodium-induced colitis mouse model, when administrated orally or through enema [196].

Essential oils have also shown promising effects on dysbiosis. For example, several essential oils were tested against species of intestinal bacteria mostly found in the human gastrointestinal tract. Overall, *Carum carvi*, *Lavandula angustifolia*, *Trachyspermum copticum* and *Citrus aurantium* var. *amara* essential oils were the most effective in inhibiting pathogens growth without affecting beneficial bacteria [197]. Another study showed that *Zanthoxylum bungeanum* essential oil showed a beneficial effect on chronic unpredictable stress-induced anxiety behaviour in rats, by restoring gut microbiota dysbiosis, namely by increasing the Sobs and Chao indexes, inhibiting Lachnospiraceae, facilitating Bacteroidales_S24-7_group, Lactobacillaceae and Prevotellaceae [198]. Essential oil emulsions of *Satureja hortensis*, *Petroselinum crispum* and *Rosmarinus officinalis* were also administered to humanized mice harbouring gut microbiota derived from patients with ischemic heart disease and Type 2 diabetes mellitus. Overall, the essential oil emulsions in mice supplemented with L-carnitine showed prebiotic effects on beneficial commensal bacteria, mainly the *Lactobacillus* genus [199]. Furthermore, a review carried out on the modulation of gut microbiota by essential oils and inorganic nanoparticles showed that essential oils are widely studied for their gastrointestinal interference, being potent modulators for gut inflammation, metabolic reactions and microbiota diversity [200].

## 4. Discussion and Future Perspectives

Studies on monoterpenes’ effect on ageing hallmarks have been carried out with many compounds presenting in their chemical structure a hydroxyl group or a phenol group that might justify the observed effects. However, an accurate chemical structure–activity relation is difficult to establish due to the multitude of study models used, concentrations tested and observed effects that hamper direct comparisons. Figure 1 shows the chemical structures of the compounds summarized in Table 1, Table 2, Table 3, Table 4, Table 5 and Table 6. Importantly, some monoterpenes like carvacrol, 1,8-cineole, limonene, perillyl alcohol and thymol have been largely assessed, showing relevant effects on different ageing hallmarks. Overall, inflammation is the hallmark mostly addressed with more than 1500 studies identified in a PubMed bibliographic search combining “monoterpenes” and “inflammation”. On the other hand, some hallmarks remain elusive, lacking studies on the effect of these compounds, namely epigenetic alterations, loss of proteostasis, deregulated nutrient sensing, stem cell exhaustion and altered intercellular communication (Figure 2).

As shown by the review herein presented, monoterpenes seem to be very promising anti-ageing compounds, but it is important to highlight that the majority of the studies assess the hallmarks individually and not in an ageing context. Indeed, several compounds are studied in a cancer setting or are performed to justify the mechanism of action associated with other properties of these compounds, namely antimicrobial or insecticidal. In these situations, monoterpenes also exerted a beneficial effect but through an enhancement of the studied hallmark, as cancer cell or microbe eradication is aimed. In addition, regarding senescence studies, the majority of the studies do not address, as recommended, at least three markers. Additional studies resorting to ageing models are, therefore, needed and could include monoterpenes with anti-inflammatory properties, as these will most likely exert a senomorphic effect by modulating components of the SASP. Moreover, essential oils have also shown very promising effects, and one cannot neglect possible synergistic effects between several compounds, including monoterpenes, present in the mixture. Therefore, further studies are still required to elucidate the real potential of monoterpenes to boost natural defences during ageing and extend healthy lifespan in the elderly.

The majority of the studies include in vitro and in vivo approaches, and, in line with this, clinical trials have also been carried out, although in much lesser extent. Interestingly, carvacrol and geraniol are the monoterpenes mostly included in clinical trials. Once again, several features of ageing hallmarks are assessed but not in an age-related context. Indeed, many of the trials are conducted in patients with lung disorders [201,202,203] or intestinal diseases like irritable bowel disease [204,205]. Furthermore, clinical translation of these compounds remains a challenge mainly due to their volatility and low water solubility and stability. Moreover, pharmacokinetic studies are sparse, and more information on the absorption, metabolism, distribution and elimination of volatile monoterpenes and essential oils is required for safe and effective use in humans. To overcome these issues, encapsulation and targeting strategies have been considered to improve bioavailability and efficacy, as reviewed elsewhere [206]. Furthermore, more large-scale and well-controlled human clinical trials taking into account the heterogeneity of ageing need to be conducted to validate the potential of these compounds and develop innovative anti-ageing strategies.

Overall, as ageing is not a disease per se, strategies that promote healthy ageing and longevity and decrease the incidence and development of ageing-related diseases are in the spotlight. In this context, monoterpenes and/or essential oils emerge as promising compounds with beneficial effects on the drivers of ageing-related diseases. These compounds can also be considered for prevention strategies, thus meeting the new developments in geriatric medicine that is gradually focusing on pre-intervention rather than post-disease treatment.

## Figures and Tables

**Figure 1 biomedicines-12-00365-f001:**
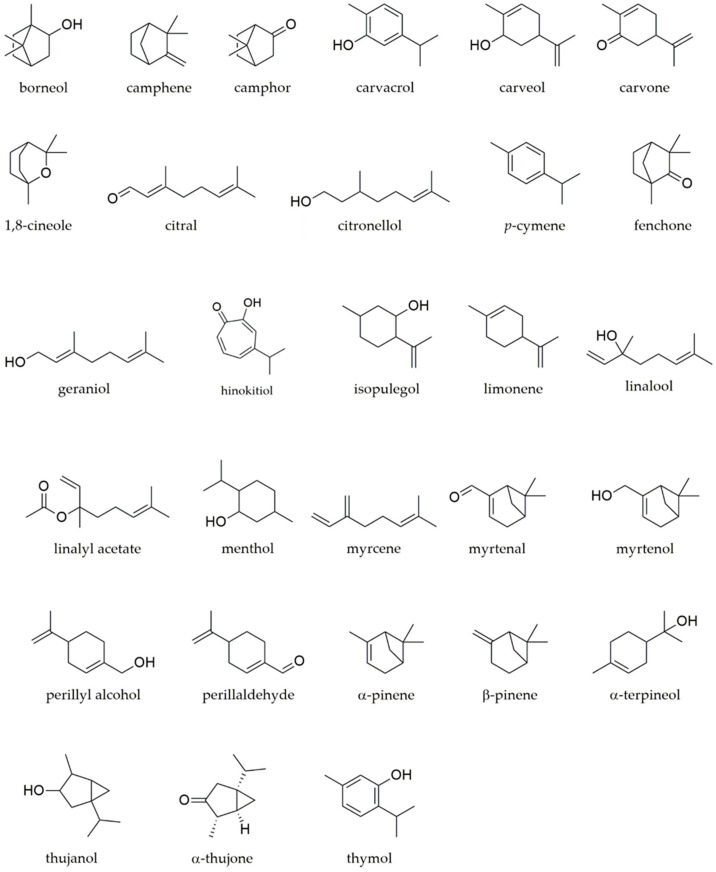
Chemical structures of monoterpenes with effects on ageing hallmarks.

**Figure 2 biomedicines-12-00365-f002:**
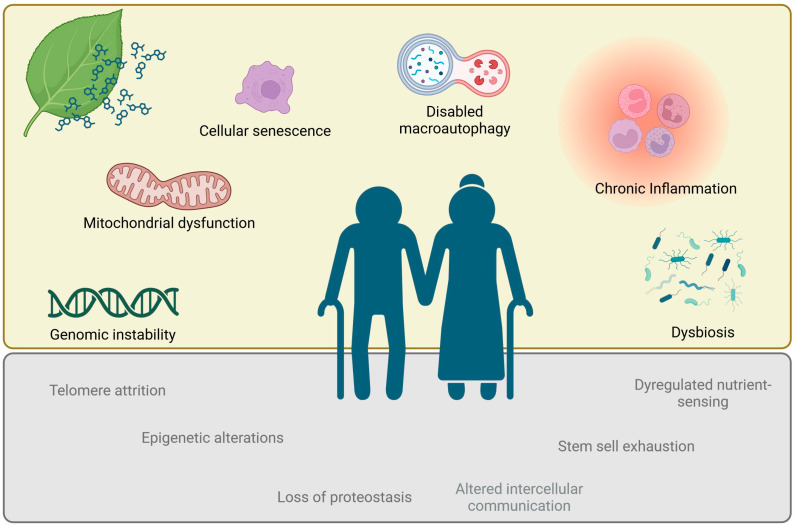
Hallmarks of ageing. Hallmarks with studies on the effect of monoterpenes are highlighted on the top part, whereas on the bottom part are shown the hallmarks lacking studies. Created with BioRender.com.

**Table 1 biomedicines-12-00365-t001:** Effect of monoterpenes on genomic instability.

Compound	Study Model	Observed Effects	Ref.
In vitro			
α-Pinene	Chinese hamster cell line—V79-Cl3 (25–50 µM; 1 h)	↑ Apoptic cells (40 and 50 µM); Multipolar or incorrectly localized spindles; ↑ Hypodiploid metaphases; ↑ Endoreduplicated cells; ↑ Chromosome breaks; ↑ Kinetochore-negative micronuclei; ↑ DNA lesions; ↑ ROS production	[23]
D-Limonene	Chinese hamster cell line—V79 (0.1–2.5 mM; 1 h)	At 2–2.5 mM: ↑ Nuclear abnormalities; ↑ Aberrant spindles; Cytokinesis failure	[24]

↑—increase; ROS—reactive oxygen species.

**Table 2 biomedicines-12-00365-t002:** Effect of monoterpenes on mitochondrial dysfunction.

Compound	Study Model	Effect	Ref.
In vitro			
1,8-Cineole	Cardiomyoblast cell line—H9c2 and primary neonatal cardiomiocytes (400 µg/mL; 24 h in cells submitted to high pressure—80 mmHg)	↓ Mitophagy; ↓ Mitochondrial fission; Restored complexes III and V protein levels	[28]
	Grass carp hepatocytes—L8824 (20 µM, in the presence of 200 nM of BPA; 24 h)	↓ ROS production; ↑ Mitochondrial membrane potential	[29]
	Grass carp hepatocytes—L8824 (20 µM, in the presence of 8 µg/mL of TPPBA; 24 h)	↓ ROS production; ↑ Mitochondrial membrane potential	[30]
	Grass carp kidney cells—CIK (20 µM, in the presence of 27.8 µg/mL of DiBP; 24 h)	↓ ROS production	[31]
Camphene	Myoblast cell line—L6 (300 µM in serum-free medium; 48 h)	Prevented mitochondrial shape alteration	[32]
Carvacrol	Neuroblastoma cell line—SH-SY5Y (100 µM; 4 h before the addition of 300 µM of H_2_O_2_; 24 h)	↑ Complexes I and V activities; ↑ ATP levels; ↑ Mitochondrial membrane potential; ↑ aconitase, α-KGDH, and SDH activities	[33]
Geraniol	Neuroblastoma cell line—SK-N-SH (60 nM; 2 h followed by 24 h with 100 nM of rotenone)	↓ ROS production; ↑ Mitochondrial membrane potential; ↑ Complex I activity; ↑ ATP production; ↑ Mitophagy of damaged mitochondria	[34]
Limonene	Skin epidermal keratinocytes cell line—HaCaT (25, 50 and 100 µM; 2 h before UVB irradation; 30 min to 24 h)	↓ ROS production; ↑ HO-1, NQO-1 and γ-GCLC protein levels; ↑ Nrf2 expression	[35]
	MG-induced damage in osteoblast—MC3T3-E1 (0.01—1 µM for 1 h prior to 48 h stimulation with 400 µM of MG)	↓ Mitochondrial superoxide levels; ↓ Cardiolipin peroxidation; ↑ ATP production; ↑ Mitochondrial membrane potential	[36]
Linalool	Hippocampal neuronal cell line— HT-22 (100 µM in the presence of 6 mM of glutamate; Respiratory capacity assays: 100 and 200 µM; 24 h)	↓ Mitochondrial fragmentation; ↓ Mitochondrial ROS production; ↑ Mitochondrial membrane potential; ↓ Mitochondrial Ca^2+^ levels; ↑ Mitochondrial respiratory capacity	[37]
Menthol	Osteosarcoma cell line—Saos2 (100 µM; 6 h)	↑ Intracellular and mitochondrial Ca^2+^ levels; ↑ Mitochondrial circularity; ↑ ATP production; ↓ Mitochondrial membrane potential; ↑ Mitochondrial ROS and cardiolipin levels; ↑ ER–mitochondria contact sites; Restored mitochondrial membrane potential in the presence of CCCP	[38]
Myrcene	Dermal fibroblasts—primary cultures (0.1, 1 and 10 μM; 24 h or 72 h after UVB irradiation)	↓ ROS production	[39]
Perillyl alcohol	Human microglial cell line—HMC-3 (100 and 200 µM in the presence of LPS/H_2_O_2_; 24 h)	↑ Mitochondrial membrane potential; ↓ Mitochondrial ROS production; ↓ H_2_O_2_ release; ↑ Parkin translocation to damaged mitochondria	[40]
	Neuroblastoma cell line—SH-SY5Y (10 and 20 µM; 1 h prior to an overnight incubation with 150 µM of 6-OHDA)	↓ Intracellular ROS production; ↑ Mitochondrial membrane potential	[41]
	Neuroblastoma cell line—SH-SY5Y (10 and 20 µM; 1 h followed by 24 h with 40 µM of β-amyloid 25–35)	↓ ROS production; ↑ Mitochondrial membrane potential (20 µM); ↑ mPTP opening	[42]
In vivo			
1,8-Cineole	Monocrotaline-induced pulmonary arterial hypertension rat model (25 mg/Kg transdermally; daily for 3 weeks)	↓ Mitophagy; ↓ Mitochondrial fission	[28]
Carvacrol	Neuropathic pain animal model (30 and 60 mg/Kg p.o.; 14 days)	↓ NO production; ↑ ATP production; ↑ NRF-1, TFAM, PGC-1α, complex I and ATP synthase C protein levels; ↓ Drp1 and Fis1 protein levels	[43]
Geraniol	ACR-induced neurotoxicity in *Drosophila melanogaster* (5 and 10 µM in culture medium; 7 days)	↓ ROS production; ↑ MTT reduction; ↑ SDH and CS activity	[44]
	STZ-induced diabetic neuropathy in rats (100 mg/Kg/day p.o.; 8 weeks)	↓ ROS production in sciatic nerve and brain; ↓ NO levels; ↑ MTT reduction; ↑ Complexes I–III, SDH, CS activity	[45]
	ACR-induced neuropathy in rats (100 mg/Kg/day p.o. and simultaneous administration of ACR, 50 mg/Kg i.p.; twice a week for 4 weeks)	↓ ROS production in sciatic nerve and brain regions; ↓ Mitochondrial ROS production in cortex and cerebellum; ↑ MTT reduction in cortex and cerebellum; ↑ Complexes I–III, complexes II and III and CS activities	[46]
Limonene	Rotenone-induced dopaminergic neurodegeneration in rats (50 mg/Kg p.o. and Rot 2.5 mg/Kg, i.p.; 5 days a week for 28 days)	↑ Mitochondrial respiratory capacity; ↑ Complex I levels	[47]
Perillyl alcohol	6-OHDA-induced Parkinson’s disease rat model (100 mg/Kg p.o.; 7-day pre- and 7-day post-surgery)	↓ Intracellular ROS production; ↑ PGC-1α mRNA and protein levels in striatum; ↑ Complexes I and IV levels; ↓ Bax and Drp1 protein levels; ↑ Nuclear accumulation of Nrf2 and PGC-1α	[48]
	MPTP-induced Parkinson’s disease rat model (100 and 200 mg/Kg/day p.o.; 14 days following 5 consecutive days of 25 mg/Kg of MPTP)	↑ ATP levels	[40]
Thymol	Grass carp (100–300 mg/Kg before a 7-day infection with *Aeromonas hydrophylla*; 68 days)	↑ Cytosolic and mitochondrial CK (100 mg/Kg); ↑ Branchial AK activity (100 and 300 mg/Kg); ↑ Branchial ATP levels; ↓ Branchial ROS production	[49]
	ISO-induced myocardial infarction rat model (7.5 mg/Kg intragastric; 7 days with ISO (100 mg/Kg), s.c., administered on days 6 and 7)	↑ Mitochondrial enzymes activity; ↓ Mitochondrial Ca^2+^ levels; ↑ ATP production; Maintained mitochondrial architecture; ↓ Mitochondrial swelling	[50]

↓—decrease; ↑—increase; 6-OHDA—6-hydroxydopamine; ACR—acrylamide; AK—Adenylate Kinase; ATP—adenosine triphosphate; Bax—BCL-2-associated protein X; BPA—bisphenol A; Ca^2+^—calcium; CCCP—carbonyl cyanide m-chlorophenyl hydrazone; CK—Creatine Kinase; CS—Citrate Synthase; DiBP—diisobutyl phthalate; GCLC—Glutamate–Cysteine Ligase Catalytic subunit; Drp1—Dynamin-related protein 1; Fis1—Mitochondrial Fission 1 protein; H_2_O_2_—hydrogen peroxide; HO-1—HemeOxygenase-1; i.p.—intraperitoneally; ISO—isoproterenol; MG—methylglyoxal; MPTP—1-methyl-4-phenyl-1,2,3,6-tetrahydropyridine; mPTP—mitochondrial permeability transition pore; mRNA—messenger ribonucleic acid; MTT—3-(4,5-dimethylthiazol-2-yl)-2,5-diphenyltetrazolium bromide; NRF-1—Nuclear Respiratory Factor 1; Nrf2 –Nuclear factor erythroid 2-Related Factor 2; NO—nitric oxide; NQO-1—NAD(P)H Quinone Dehydrogenase 1; LPS—lipopolysaccharide; p.o.—per os (orally); PGC-1α—Peroxisome proliferator-activated receptor-gamma Coactivator-1alpha; ROS—reactive oxygen species; ROT—rotenone; s.c. –subcutaneously; SDH—Succinate Dehydrogenase; TBBPA—tetrabromobisphenol A; TFAM—Mitochondrial Transcription Factor A; UVB—Ultraviolet B; α-KGDH—alpha-KetoGlutarate DeHydrogenase.

**Table 3 biomedicines-12-00365-t003:** Effect of monoterpenes on cellular senescence.

Compound	Study Model	Effect	Ref.
In vitro			
Camphor	Primary cultures of dermal fibroblasts (65, 130 and 260 µM; 24 h and 260 µM; 6, 12 or 24 h)	↓ Elastase activity; ↑ Elastin and collagen productions; ↓ SA-β-galactosidase activity (in the presence of H_2_O_2_)	[69]
Hinokitiol	Skin fibroblast cell line—966SK following UVB irradiation (4, 8 and 12 µM; 24 h)	↓ Secreted MMP-1 levels; ↓ MMP-1 and MMP-3 mRNA levels; ↑ Procollagen mRNA levels	[70]
D-Limonene	Skin epidermal keratinocytes cell line—HaCaT (25, 50 and 100 µM for 2 h before UVB irradation; 30 min to 24 h)	↓ α-MSH intracellular levels; ↓ POMC mRNA; ↓ Phosphorylated levels of p53; Maintained skin barrier function	[35]
Myrcene	Normal human dermal fibroblasts (0.1, 1 and 10 μM; 1, 4, 24 or 72 h after UVB irradiation)	↓ ROS production; ↓ MMP-1 and MMP-3 secretion; ↓ IL-6 secretion; ↑ Procollagen-1 and TGF-1β secretion; ↓ MMP-1 mRNA levels; ↑ Procollagen mRNA levels; ↓ MAPK activation; ↓ AP-1 activation	[39]
α-Pinene	Skin epidermal keratinocytes cell line—HaCaT (30 µM; 30 min before UVA irradiation, then cultured for 24 h)	↓ Single-strand DNA damage; ↓ Pyrimidine dimers formation; ↑ NER pathway-associated proteins expression; ↑ p53 and p21 protein levels	[71]
In vivo			
Camphor	UV-induced wrinkle formation in mice (26 and 52 mM, topical; for 2 weeks after 4 weeks of UV irradiation)	↑ Collagen 1, collagen II and elastin production; ↓ MMP-1 protein levels; ↓ Epidermis and subcutaneous fat layer thickness	[69]
α-Pinene	UVA-induced photoageing mice model (100 mg/Kg, topical; 1 h prior to irradiation)	↑ Collagen staining; ↓ MMP-2 staining; ↓ MMP-9 and MMP-13 mRNA expression; ↓ MMP-1 and MMP-9 protein levels	[72]

↓—decrease; ↑—increase; AP-1—Activator Protein 1; IL—InterLeukin; MAPK—Mitogen-Activated Protein Kinase; MMP—Matrix MetalloProteinase; NER—Nucleotide excision repair; POMC—pro-opiomelanocortin; SA—senescence-associated; TGF-1β—Transforming Growth Factor 1β; UV—Ultraviolet; UVA—Ultraviolet A; UVB—Ultraviolet B; α-MSH—α-Melanocyte-Stimulating Hormone.

**Table 4 biomedicines-12-00365-t004:** Effect of monoterpenes on autophagy.

Compound	Study Model	Effect	Ref.
In vitro			
Carvacrol	Cardiomyoblasts—H9c2 cells (2.5, 5 and 10 µg/mL; 6 h followed by 1 µg/mL of LPS; 18 h)	↑ Beclin-1 protein levels; ↓ p62 protein levels	[95]
1,8-Cineole	Grass carp hepatocytes—L8824 (20 µM and 200 nM of BPA; 24 h)	↑ LC3B fluorescence; ↓ p62 fluorescence; ↑ LC3, Atg5 and beclin-1 mRNA and protein levels; ↓ p62 mRNA and protein levels	[29]
	Human umbilical vein endothelial cells—HUVEC (5 and 50 µM; 1.5 h followed by 300 µM of L-NAME; 24 h)	↓ LC3-II/LC3-I ratio; ↑ p62 mRNA and protein levels; ↓ LC3 fluorescence; ↑ PI3K and mTOR phosphorylated levels	[96]
	Human bronchial epithelial cell line—BEAS-2B (6.25–200 µM; 2 h, before 4 h with CSE)	↓ LC3-II/LC3-I ratio	[97]
	*Ctenopharyngodon idellus* kidney cells (20 µM and 27.8 µg/mL of DiBP; 24 h)	↑ Autophagosome formation; ↑ Atg5, beclin-1 and LC3 mRNA levels; ↓ p62 mRNA and protein levels; ↑ LC3 and beclin-1 protein levels	[31]
Hinokitiol	Neuroblastoma cell line—SK-N-SH (2, 4 and 8 µM; 6 h followed by 100 µM of PrP; 6 h)	↑ LC3-II/LC3-I ratio; ↑ LC3 puncta; ↑ Autophagosome formation; ↑ p62 mRNA and protein levels	[98]
	Cardiomyocyte cell line—AC16 (20 µM; 30 min followed by 1 mM of H_2_O_2_; 2 h)	↑ Phosphorylated form of mTOR and p62 protein levels; ↓ Beclin-1 protein levels and LC3-II/LC3-I ratio; ↓ LC3-positive cells	[99]
Menthol	Bovine mammary epithelial cells (200 µM; 1 h followed by 5 µg/mL of LPS; 12 h)	↓ p62 protein levels; ↑ LC3-II protein levels; ↑ Autophagosomes and autophagolysosome formation	[100]
Thymol	Mouse liver cell line—AML12 (100 µM in the presence of 100 mM of EtOH; 48 h)	↓ LC3-II and p62 protein levels; ↓ LC3-II/LC3-I ratio	[101]
In vivo			
Borneol	Cerebral ischemia/reperfusion rat model (0.16 mg/Kg/day p.o.; 7 days)	Cortex: ↑ ULK1 protein levels and LC3-II/LC3-I ratio Hippocampus: ↑ ULK1 protein levels and LC3-II/LC3-I ratio; ↓ mTOR	[102]
Carvacrol	LPS-induced cardiac dysfunction mice model (50 and 100 mg/Kg i.p.; 2 h prior to 10 mg/Kg of LPS; 12 h)	↑ Beclin-1 protein levels; ↓ p62 protein levels	[95]
	Chronic constriction injury of sciatic-nerve-induced neuropathic pain in rats (30 and 60 mg/Kg/day p.o.; 14 days)	↑ Beclin-1, LC3-II, Atg7 and Atg16 protein levels; ↓ p62 protein levels and immunostaining	[43]
1,8-Cineole	L-NAME-induced hypertension rat model (20 and 40 mg/Kg/day, gavage on weeks 5 to 8 during an 8-week administration of L-NAME 40 mg/Kg/day)	↓ LC3-II/LC3-I ratio; ↑ p62 mRNA and protein levels; ↓ LC3 immunohistochemical staining in rat blood vessels; ↑ PI3K and mTOR phosphorylated levels	[96]
Citronellol	Rotenone-induced Parkinson’s disease rat model (25 mg/Kg p.o.; 30 min before 2.5 mg/Kg i.p. of Rot; 4 weeks)	↓ LC3 and p62 protein levels	[103]
Geraniol	ISO-induced myocardial infarction (100 and 200 mg/Kg/day p.o. followed by 100 mg/Kg s.c. of ISO on days 13 and 14)	↑ PI3K, Akt and mTOR mRNA levels; ↑ Phosphorylated levels of PI3K, Akt and mTOR	[104]
Myrcene	Rotenone-induced Parkinson’s disease rat model (25 mg/Kg p.o.; 30 min before 2.5 mg/Kg i.p. of Rot; 5 days a week for 28 days)	↓ Beclin-1, p62 and LC3B protein levels; ↑ p-mTOR/mTOR ratio	[105]
Thymol	LPS-induced liver inflammation mouse model (80 mg/Kg/day, gavage, for 34 days followed by 10 mg/Kg, i.p. of LPS; 4 h)	↓ p62 mRNA levels; ↑ Atg7 mRNA levels; ↑ phosphorylated levels of AMPK	[106]

↓—decrease; ↑—increase; Akt—Protein kinase B; AMPK—5′Adenosine Monophosphate-activated Protein Kinase; Atg—autophagy-related; BPA—bisphenol A; CSE—cigarette smoke extract; DiBP—diisobutyl phthalate; EtOH—ethanol; H_2_O_2_—hydrogen peroxide; i.p.—intraperitoneally; ISO—isoproterenol; L—NAME—N(ω)-nitro-L-arginine methyl ester; LC3—Microtubule-associated protein Light Chain 3; LPS—lipopolysaccharide; mRNA—messenger ribonucleic acid; mTOR—mammalian Target Of Rapamycin; p.o.—per os (orally); PrP—Prion Protein; p62/SQSTM1—Sequestosome-1; PI3K—PhosphoInositide 3-Kinases; Rot—rotenone; ULK1—Unc-51-Like Kinase 1.

**Table 5 biomedicines-12-00365-t005:** Effect of monoterpenes on inflammation.

Compound	Study Model	Observed Effects	Ref.
In vitro			
Carvacrol	Human umbilical vein endothelial cells—HUVEC (10 mg/Kg; 24 h in HG medium)	↓ IKK, NALP3, NF-κB and TLR4 mRNA and protein levels	[118]
(S)-(+)-Carvone	Mouse leukemic cell line—RAW 264.7 (666 µM; 5 min and 1 h, before 1 µg/mL of LPS)	↓ p-JNK 1; ↓ Ac-p65; ↓ IκB-α resynthesis; ↑ SIRT 1 activity (in chemico)	[119]
(R)-(-)-Carvone	Mouse leukemic cell line—RAW 264.7 (666 µM; 5 min, 1 h and 18 h before 1 µg/mL of LPS)	↓ p-JNK 1; ↓ Ac-p65 (tendency); ↓ IκB-α resynthesis; ↑ Nrf2 translocation (tendency); ↑ HO-1 protein level (tendency)	[120]
1,8-Cineole	Human bronchial epithelial cell line—BEAS-2B (6.25–200 µM; 2 h, before 4 h with CSE)	↑ Nrf2 nuclear translocation; ↓ ROS and IL-6 levels	[97]
	Mouse leukemic cell line—RAW 264.7 (0.1, 1, 5 and 10 µM; 20 min, before 0.5 mg/mL of MSU for 4 h)	↑ Nrf2 protein levels; ↓ TNF-α, IL-6, CXCL1 and CXCL2 mRNA expression	[121]
Citral	Porcine jejunal epithelial cell line—IPEC-J2 (PGN-stimulated cells)	↓ Cytokine expression; ↓ TLR2 protein levels; ↓ TLR2/NF-κB signaling pathway	[122]
	Human peripheral blood mononuclear cells—PBMC (4, 2 and 1%; 30 min before and after innoculation with *S. aureus* for 6 h—TI/IT, respectively)	TI: ↓ IL-1β, IL-6, IL-12p70, IL-23, IFN-γ and TNF-α levels IT: ↓ IL-1β, TNF-α and IL-6 (only 4%)	[123]
Geraniol	Microglial cell line—BV-2 (200-, 500- and 1000-fold concentrations; 24 h pretreatment with geraniol followed by 24 h with 1 µg/mL of LPS. Pretreatment with LPS; 24 h, followed by 24 h with geraniol, co-stimulation with LPS and geraniol for 24 h)	Geraniol pretreatment: ↓ TNF-α and IL-6 mRNA and secretion; ↓ p50 and p-C/EBP-β chromatin-bound protein levels LPS pretreatment: ↓ TNF-α and IL-6 mRNA and secretion; ↓ p50 and p-C/EBP-β chromatin-bound protein levels Co-treatment: ↓ TNF-α and IL-6 mRNA and secretion; ↓ p50 and p65 chromatin-bound protein levels	[124]
	Primary cultures of macrophages (5, 25 and 50 µM; 12 h followed by 1 µg/mL of LPS for 24 h)	↓ IL-1β protein levels; ↓ IL-1β and NLRP3 mRNA expression; ↓ TNF-α mRNA expression; ↑ IL-4 and IL-10 mRNA expressions	[125]
	Human umbilical vein endothelial cells—HUVEC (0–100 µmol/mL; 2 h before 100 µg/mL of OxLDL for 72 h)	↓ TNF-α, IL-6, IL-1β, ICAM-1, VCAM-1 and TGF-β protein levels; ↓ IL-6, VCAM-1 and ICAM-1 mRNA levels; ↓ p-IκBα and p-p65 protein levels	[126]
Hinokitiol	Mouse leukemic cell line—RAW 264.7 (5, 10 and 20 µg/mL; 1 h, before 100 ng/mL of LPS for 3 h)	↓ IL-6, IL-1β, TNF-α and NLRP3 mRNA expression	[127]
Linalool	Microglial cell line—BV-2 (200-, 500- and 1000-fold concentrations; 24 h pretreatment with linalool followed by 24 h with 1 µg/mL of LPS. Pretreatment with LPS; 24 h, followed by 24 h with linalool, co-stimulation with LPS and linalool for 24 h)	Linalool pretreatment: ↓ TNF-α and IL-6 mRNA and secretion; ↓ p50, p65 and p-C/EBP-β chromatin-bound protein levels LPS pretreatment: ↓ TNF-α mRNA and secretion; ↓ IL-6 secretion; ↓ p50 and p-C/EBP-β chromatin-bound protein levels Co-treatment: ↓ TNF-α secretion	[124]
Menthol	Microglial cell line—BV-2 (5–40 µM; 2 h, before 1 µg/mL of LPS for 2 or 12 h)	↓ iNOS and COX-2 mRNA and protein levels; ↓ IL-1β, IL-6 and TNF-α mRNA and protein expression; ↓ p-p65/p65, p-Akt/Akt, p-ERK/ERK, p-JNK/JNK and p-p38/p38 ratios	[128]
Myrcene	Dermal fibroblasts—primary cultures (0.1, 1 and 10 μM; 24 h or 72 h after UVB irradiation)	↓ IL-6 release	[39]
Perillyl alcohol	Human keratinocytes cell line—HaCaT (50 and 100 µM; 2 h prior to 1 µg/mL of LPS for 22 h)	↓ p-p65 and p-STAT3 nuclear accumulation	[129]
	Human embryonic kidney—HEK293 (25 and 50 µM; 24 h following 30 ng/mL of TNF-α for 2 h)	↓ NF-κB activation; ↓ p-NF-κB and p-IκBα immunoreactivity	[130]
Perillaldehyde	Human corneal epithelial cell line—HCEC (0.6 mM; 4 h, followed by 8 h stimulation with *A. fumigatus*)	↓ IL-1β, IL-6, TNF-α and IL-8 mRNA expression; ↓ IL-6, TNF-α and IL-8 secretion; ↑ Nrf2 and HO-1 mRNA and protein expression; ↓ Dectin-1 mRNA and protein levels	[131]
α-Pinene	Skin epidermal keratinocytes cell line—HaCaT (30 µM; 30 min before UVA irradiation, cultured for an additional 24 h)	↓ NF-κB, IL-6 and TNF-α protein levels	[71]
Thujanol	Microglial cell line—BV-2 (200-, 500- and 1000-fold concentrations; 24 h pretreatment followed by 24 h with 1 µg/mL of LPS. Pretreatment with LPS; 24 h, followed by 24 h with thujanol, co-stimulation with LPS and thujanol for 24 h)	Thujanol pretreatment: ↓ TNF-α and mRNA and secretion; LPS pretreatment: ↓ TNF-α secretion; ↓ p50, p65 and p-C/EBP-β chromatin-bound protein levels. Co-treatment: ↓ p50 chromatin-bound protein levels	[124]
Thymol	Mouse leukemic cell line—RAW 264.7 (30–150 µM; 2 h before 1 µg/mL of LPS for 24 h)	↓ NO production; ↓ TNF-α and IL-6 release (120 µM); ↓ ROS production; ↓ COX-2 protein expression; ↓ p65 nuclear translocation	[132]
In vivo			
Borneol	Cerulein-induced acute pancreatitis mice model (100 and 300 mg/Kg, p.o.; 7 days, on day 7, 6 injections of cerulein (50 µg/Kg, i.p.) were given 1 h apart, sacrifice 6 h after last injection)	↓ NO, IL-1β and IL-6 levels; ↓ MPO activity; ↓ iNOS, IL-1β, p-NF-κB, TNF-α and IL-6 protein levels; ↓ Inflammatory cells infiltration	[133]
Carvacrol	Hyperuricemia-induced inflammation rat model (20 and 50 mg/Kg i.p.; 7 days, 1 h after PO administration)	↓ Monourate crystals; ↓ Lymphocyte infiltration; ↓ TNFα and NRLP3 levels; ↓ TNF-α and p-NF-κB staining	[134]
	CdCl_2_-induced neurotoxicity rat model (25 and 50 mg/mL wth 25 mg/Kg p.o. of CdCl_2_; daily for 7 days)	↓ NF-κB release; ↓ Brain COX-2, MPO and PGE_2_ levels; ↓ Brain nNOS, iNOS, GFAP and MAO levels; ↓ TNF-α, IL-1β, MMP-9 and MMP-13 mRNA levels	[135]
	CdCl_2_-induced lung injury rat model (25 and 50 mg/Kg, p.o.; 30 min after 25 mg/Kg, p.o. of CdCl_2_ for 7 days)	↓ Lung NF-κB, iNOS, COX-2, MPO and PGE_2_ levels; ↓ TNF-α, IL-1β, MMP-2 and MMP-9 mRNA expression	[136]
	T2DM db/db mice model (5 and 10 mg/Kg, gavage; 6 weeks)	↓ IL-1β, IL-6, IL-18 and TNF-α secretion; ↓ IKK, NALP3, NF-κB and TLR4 mRNA and protein levels	[118]
	FM1-induced viral infection mice model (50 mg/Kg, intranasally; 5 days post viral infection)	↑ % of Treg cells; ↓ Th1/Th2 and Th17/Treg ratio; ↓ IFN-γ, IL-2, IL-4, IL-12, TNF-α, IL-1, IL-10 and IL-6 secretion; ↓ RIG-I, MyD88, NF-κB mRNA and protein levels	[137]
	LPS-induced memory impairment rat model (25, 50 and 100 mg/Kg, i.p.; until 21 post-LPS sacrifice at day 28 post-LPS)	↓ IL-1β, IL-6, TNF-α, COX-2 and NF-κB levels; ↓ iNOS, TLR4 and BDNF mRNA expression	[138]
	ACR-induced liver damage rat model (50 mg/Kg, i.p., followed by 20 mg/Kg, p.o. of ACR; 30 days)	↓ TNF-α, IL-1β and NF-κB protein levels	[139]
Carveol	PTZ-kindled epileptic rat model (10 and 20 mg/Kg 30 min before PTZ; repeated every 48 h for 15 days)	↓ TNF-α, p-p65 and COX-2 protein levels	[140]
S-Carvone	Stress-induced liver damage rat model (20 mg/Kg, gavage, with restraint for 6 h; daily for 21 days)	↓ TNF-α, IL-6, IL-1β and NF-κB mRNA expression; ↓ Inflammatory cells infiltration	[141]
	Cerebral I/R-induced neuroinflammation rat model (10 and 20 mg/Kg, i.p.; 15 min before reperfusion daily for 15 days)	↓ Serum and brain IL-1β and TNF-α levels; ↓ IL-6 and IL-4 levels; ↑ IL-10 levels; ↓ NLRP3, ASC, TLR4, IL-1β, TNF-α mRNA levels	[142]
	Hepatic I/R-induced injury rat model (25 and 50 mg/Kg, gavage; 3 weeks before I/R)	↓ HMGB1, TLR4, NF-κB and NLRP3 mRNA expression; ↓ TLR4 and NF-κB immunoreactivity; ↓ ICAM-1, MPO, IL-1β, IL-6 and TNF-α protein levels; ↑ IL-10 protein levels	[143]
1,8-Cineole	SAH-induced early brain injury rat model (100 mg/Kg, i.p.; 1 h before SAH and 30 min after)	↓ Iba-1 and p65 protein levels; ↓ TNF-α, IL-1β and IL-6 mRNA levels	[144]
	MSU-induced gout arthritis mice model (30–600 mg/Kg, i.p.; 1 h before and 5, 23 and 47 h after 0.5 mg/20 µL of MSU, intra-articular)	↓ Ankle edema; ↓ cell infiltration; ↓ MPO activity; ↓ NLRP3 and IL-1β mRNA and protein levels	[121]
	DSS-induced ulcerative colitis mice model (100 and 200 mg/Kg with 2% DSS; 8 days)	↓ IL-6, IL-1β, TNF-α, IL-17A protein and mRNA levels; ↓ iNOS and COX-2 mRNA and protein levels; ↓ p-p65/p65 ratio	[145]
Citral	*C. sakazakii*-induced intestinal inflammation in newborn mice (0.54 mg/mL, gavage; once per day for 8 days, starting on day 3 postnatal, at day 7, *C. sakazakii* p.o. was given, sacrifice at day 10)	↓ IL-1β, TNF-α, PAF receptor, IL-6, IFN-γ, NF-κB p65 and iNOS mRNA levels; ↓ IL-6 and TNF-α levels; ↓ NF-κB p65 protein levels; ↑ IκBα protein levels	[146]
*p*-Cymene	TNBS-induced intestinal inflammation rat model (25, 50, 100 and 200 mg/Kg, p.o.; 48 h, 24 h and 1 h before and 24 h after 10 mg/animal TNBS)	↓ Inflammatory lesions; ↓ MPO activity; ↓ IL-1β and TNF-α secretion; ↑ IL-10 secretion;↓ COX-2, IFN-γ, iNOS, p65 and SOCS3 mRNA expression	[147]
Fenchone	FCA-induced arthritis rat model (100, 200 and 400 mg/Kg; daily for 28 days post FCA injection)	↓ Paw volume and arthritis severity; ↓ iNOS, IL-17, COX-2, IL-1β, TNF-α and IL-6 mRNA levels; ↑ IL-10 levels; ↓ NO and PGE_2_ production	[148]
Geraniol	ISO-induced myocardial infarction rat model (100 mg/Kg/day p.o.; 14 days, 85 mg/Kg of ISO i.p., in the last 2 days)	↓ TNF-α, IL-6 and NF-κB protein levels; ↓ MMP-9 and MMP-2 mRNA levels	[149]
	*Mycoplasma pneumoniae*-induced pneumonia mice model (20 mg/Kg; 3 days, after 2-day infection by *M. pneumoniae*)	↓ IL-1, IL-6, IL-8, TNF-α and TGF levels; ↓ ERK1/2 and NF-κB mRNA expression	[150]
	Cyclosporine A-induced renal injury rat model (50, 100 and 200 mg/Kg, intragastric, 1 h prior to 30 mg/Kg of cyclosporine; daily for 28 days)	↓ NF-κB mRNA levels; ↓ Renal IL-18 and ICAM-1 levels; ↓ Renal TGF-β levels; ↓ MMP-9 mRNA expression	[151]
	FCA-induced arthritis rat model (25, 50 and 100 mg/Kg, i.p.; days 7 to 35 post FCA injection, on alternate days)	↓ Paw edema; ↓ Arthritis severity; ↓ NF-κB, IL-1β, TNF-α, COX-2, mPGES-1, PTGDS and MMP-1 mRNA levels	[152]
Hinokitiol	Tooth-ligation-induced periodontitis mice model (2 mg/mL injected into the palatal gingiva; daily for 7 days)	Prevented bone loss; ↓ IL-6, IL-1β, TNF-α and NLRP3 mRNA expression;	[127]
Limonene	Gentamycin-induced acute kidney injury rat model (100 mg/Kg p.o.; 1 h before 100 mg/Kg, i.p., of gentamycin; daily for 12 days)	↓ NO production; ↓ Renal TNF-α, IL-6 and NF-κB mRNA levels; ↓ TNF-α immunoreactivity	[153]
	HFD-induced atherosclerosis in diabetic rat model (200 mg/Kg, gavage on day 30, under HFD; for 8 weeks)	↓ TNF-α and IL-6 protein levels; ↑ IL-10 protein levels; ↑ p-AMPK/APMK ratio; ↓ p-p65/p65 ratio	[154]
Menthol	LPS-induced Parkinson’s disease rat model (10 and 20 mg/Kg, gavage; 28 days post LPS injection)	↓ Iba-1-positive cells; ↓ OX-42 protein levels; ↓ iNOS and COX-2 protein levels; ↓ IL-1β, IL-6, TNF-α, COX-2 and iNOS mRNA expression; ↓ p-p65/p65, p-Akt/Akt, p-ERK/ERK, p-JNK/JNK and p-p38/p38 ratios	[128]
	LPS-induced neuroinflammation mice model (26 and 52 mg/Kg, p.o. with 0.2 mg/Kg, i.p., of LPS; daily for 12 days)	↓ Iba1-positive microglia; ↓ TNF-α, IL-1β and IL-6 protein levels; ↑ BDNF, TrkB protein levels and p-CREB/CREB ratio	[155]
Myrcene	STZ-induced diabetic rat (25 mg/Kg, 72 h after diabetes induction; for 45 days)	↓ TNF-α and IL-6 serum levels; ↓ TGF-1β, TNF-α and NF-κB levels; ↓ MCP-1, VCAM-1 and ICAM-1 levels; ↑ MMP-2 protein levels; ↓ TIMP1- protein levels	[156]
	ADX-induced renal inflammation rat mode (100 mg/Kg; for 14 days post-surgery)	↓ IL-1β, IL-6, TNF-α, p65, and IL-4 levels; ↑ IFN-γ and IL-10 levels; ↓ iNOS and COX-2 protein levels	[157]
Myrtenal	STZ-induced diabetes rat model (80 mg/Kg, p.o.; daily for 4 weeks)	↓ TNF-α, IL-6 and NF-κB liver and pancreas mRNA and protein levels	[158]
Myrtenol	STZ-induced diabetes in pregnant rat model (50 mg/Kg, p.o.; daily for 2 weeks, starting on gestational day 4)	↓ TLR4, MyD88, p65 and NRLP3 mRNA levels; ↓ TNF-α and IL-1β production	[159]
Perillyl alcohol	Imiquimod-induced psoriasis-like skin inflammation (100 and 200 mg/Kg, topical; days 3 to 7 post 62.5 mg/Kg imiquimod	↓ Psoriasis severity (scaling, epidermal thickness, erythema); ↓ Skin NO, IL-1β, IL-6, IL-12/IL-23p40 and TNF-α levels; ↓ Skin TLR7, TLR8, IL-23, IL-17 and IL-22 mRNA expression; ↓ Skin iNOS, IL-17A, TNF-α, p-p65, COX-2, IL-22 and IL-10 protein levels	[129]
	DSS-induced ulcerative colitis mice model (100 and 200 mg/Kg, p.o.; from day 15 to day 28, chronic restraint stress days 1 to 28 and 2.5% DSS days 8 to 14)	↓ NO, IL-1β, IL-6, TNF-α release; ↓ IL-1β, TNF-α, TLR4 and NF-κB mRNA expression; ↓ MPO activity; ↓ IL-6, p-IκBα and TNF-α protein levels; ↓ p-NF-κB/NF-κB ratio	[130]
α-Pinene	UVA-induced photoageing mice model (100 mg/Kg, topical; 1 h prior to irradiation)	↓ COX-2, NF-κB, iNOS, VEGF and CD34 expression in mouse skin; ↓ COX-2, iNOS, VEGF protein levels; ↓ Nuclear translocation of NF-κB	[72]
	ISO-induced myocardial infarction rat model (50 mg/Kg, p.o.; 21 days with 85 mg/Kg, s.c., of ISO, on days 20 and 21)	↓ TNF-α and IL-6 secretion; ↓ TNF-α, IL-6 and NF-κB protein levels	[160]
α-Terpineol	DSS-induced ulcerative colitis (50 mg/Kg, p.o.; 14 days, DSS in water from days 7 to 14)	↓ NO and MPO content; ↓ Mast cell infiltration; ↓ NF-κB p65, COX-2 and iNOS immunoreactivity;	[161]
Thymol	IMID-induced brain damage rat model (30 mg/Kg, p.o., 1 h before 22.5 mg/Kg, p.o., of IMID; 56 days)	↓ NO and MPO content; ↓ NF-κB immunoreactivity	[162]
	Bleomycin-induced pulmonary fibrosis mice model (50 and 100 mg/Kg, p.o., with 15 mg/Kg, i.p., of bleomycin; daily for 4 weeks)	↓ Inflammatory infiltrate and edema; ↓ TNF-α, IL-1β, IL-6 and NF-κB protein levels	[163]

↓—decrease; ↑—increase; Ac—acetylated; ACR—acrylamide; ADX—adrenalectomy; Akt—Protein kinase B; AMPK—5′Adenosine Monophosphate-activated Protein Kinase; ASC—Apoptosis-associated speck-like protein containing a caspase recruitment domain; BDNF—Brain-Derived Neurotrophic Factor; CD35—cluster of differentiation 35; C/EBP-β—CCAAT/Enhancer-Binding Protein beta; CdCl_2_—cadmium chloride; COX—Cyclooxygenase; CREB—cAMP Response Element-Binding protein; CSE—cigarette smoke extract; CXCL—chemokine (C-X-C motif) ligand; DSS—dextran sulphate sodium; ERK—Extracellular signal-Regulated Kinases; FCA—Freund’s complete adjuvant; GFAP—Glial Fibrillary Acidic Protein; HFD—high-fat diet; HG—high glucose; HMGB1—High-Mobility Group Box 1 protein; HO-1—HemeOxygenase-1; I/R –ischemia/reperfusion; Iba-1—ionized calcium-binding adaptor molecule 1; ICAM—InterCellular Adhesion Molecule; IKK –Nuclear Factor kappa B Inhibitor kinase; IL—InterLeukin; IMID—imidacloprid; INF-γ—Interferon gamma; iNOS—inducible Nitric Oxide synthase; ISO—isoproterenol; i.p.—intraperitoneally; IκBα –Nuclear Factor kappa B Inhibitor; JNK—c-Jun N-terminal Kinase; LPS—lipopolysaccharide; MAO—MonoAmine Oxidases; MCP-1—Monocyte Chemoattract Protein-1; MMP—Matrix MetalloProteinase; mPGES-1—microsomal ProstaGlandin E synthase-1; MPO—MyeloPerOxidase; mRNA—messenger ribonucleic acid; MSU—monosodium urate; MyD88—Myeloid Differentiation primary response 88; NALP3—NACHT, LRR and PYD domains-containing protein 3; NF-κB—Nuclear Factor kappa B; nNOS—neuronal Nitric Oxide synthase; NO –nitric oxide; Nrf2—Nuclear factor erythroid 2-Related Factor 2; NRLP3—NLR family pyrin domain-containing 3; oxLDL—oxidized Low-Density Lipoprotein; OX-42—CD11a/b; p50—nuclear factor NF-kappa-B p50 subunit; p65—nuclear factor NF-kappa-B p65 subunit; PAF—Platelet-Activating Factor; PGE 2—ProstaGlandin E 2; PGN—Peptidoglycan; p.o.—per os (orally); PO –Potassium monooxonate; PTGDS—ProstaGlandin D2 synthase; PTZ—Pentylenetetrazole; RIG-I—Retinoic acid-Inducible Gene I; ROS—reactive oxygen species; SAH –subarachnoid heamorrhage; SIRT1—Sirtuin-1; SOCS3—Suppressor Of Cytokine Signaling 3; STAT3—Signal Transducer and Activator of Transcription 3; STZ—Streptozotocin; T2DM—Type 2 diabetes mellitus; TGF-β—Transforming Growth Factor beta; TIMP—Tissue Inhibitors of MetalloProteinases: TLR—Toll-Like Receptor; TNBS—Trinitrobenzenesulfonic acid; TNF—Tumor Necrosis Factor; TrkB—Tropomyosin receptor Kinase B; UVA—Ultraviolet A; UVB—Ultraviolet B; VCAM—Vascular Cell Adhesion Molecule; VEGF—Vascular Endothelial Growth Factor.

**Table 6 biomedicines-12-00365-t006:** Effect of monoterpenes on dysbiosis.

Compound	Study Model	Observed Effects	Ref.
Geraniol	DSS-induced colitis mouse model (30 and 120 mg/Kg, p.o., for 17 days (days 8–24), DSS (1.5%) was given in tap water for 7 days (days 17–23); total time 37 days	Maintained a microbiota composition similar to healthy mice (120 mg/Kg); ↑ *Bacteriodetes* population; ↑ Lactobacillaceae population higher than control (120 mg/Kg, day 25)	[196]
	DSS-induced colitis mouse model (120 mg/Kg, enema administration on days 19, 21, 23 and 25, DSS (1.5%) was given in tap water for 7 days (days 17–23); total time 37 days	Maintained a microbiota composition similar to healthy mice; ↑ *Bacteriodetes* population; ↑ Lactobacillaceae population higher than control (day 25)	[196]

↑—increase; DSS—dextran sulphate sodium; p.o.—per os (orally).

## Data Availability

Data are contained within the article.

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
