# Peer review of "Plant Monoterpenes and Essential Oils as Potential Anti-Ageing Agents: Insights from Preclinical Data"

_biomedicines, 2024, doi:10.3390/biomedicines12020365_

Round 1

Reviewer 1 Report

Comments and Suggestions for Authors

The article presents a comprehensive study on the potential bioactive volatile molecules, mainly monoterpenes, with many studies referring to their anti-ageing potential. The authors provide an updated systematization of the bioactive potential of volatile monoterpenes on recently proposed ageing hallmarks, and highlight the main mechanisms of action already identified, as well as possible chemical entity-activity relations.

Before recommending this article for publication some suggestions are needed as follow:

-          Line 85: “those form” should be those from.

-          Line 100: “can found” should be can be found.

-          Check description for Table 6.

-          Line 602: “trough” should be through.

-          For a better correlation and highlight, maybe it would be better to write the chemical structure for the mentioned compounds in Tables 1-6.

Author Response

The article presents a comprehensive study on the potential bioactive volatile molecules, mainly monoterpenes, with many studies referring to their anti-ageing potential. The authors provide an updated systematization of the bioactive potential of volatile monoterpenes on recently proposed ageing hallmarks, and highlight the main mechanisms of action already identified, as well as possible chemical entity-activity relations.

Before recommending this article for publication some suggestions are needed as follow:

-  Line 85: “those form” should be those from.

-  Line 100: “can found” should be can be found.

-  Check description for Table 6.

-  Line 602: “trough” should be through.

Thank you so much for the detecting these mistakes. We have corrected accordingly.

- For a better correlation and highlight, maybe it would be better to write the chemical structure for the mentioned compounds in Tables 1-6.

We appreciate the suggestion. However, we would like to highlight that several compounds are referred to in more than one the table, so the inclusion of their chemical structure would be quite repetitive along the manuscript. For this reason, we chose to compile the chemical structures in figure 1.

Reviewer 2 Report

Comments and Suggestions for Authors

The paper ‘Are aromatic plants and their volatile monoterpenes potential 2 anti-ageing agents?’ presents an overview of recent articles on the possible use of selected plants and their active ingredients, especially volatile monoterpenes as potential anti-ageing agents. At the beginning of the article, the authors included information on the mechanisms of aging. In the following section, they described the possibility of using selected monoterpenes to delay the aging process by quoting the results of a number of recent studies focusing on both the compounds with the expected effects, as well as presenting their mechanism of action. Moreover, the chemical formulas of the compounds with anti-ageing properties as well as short information on isolation techniques from plants  are also included.  The article should be published because of the important aspects raised in it related, as mentioned by the authors, to the extension of life and the increasing number of elderly people suffering from various diseases related to, among other things, ageing.

However, the citation of literature in the text should be slightly corrected in accordance with editorial requirements (lines 173, 176, 179, 185, Table 4, 514, 563, 568, 575)

Author Response

agents?’ presents an overview of recent articles on the possible use of selected plants and their active ingredients, especially volatile monoterpenes as potential anti-ageing agents. At the beginning of the article, the authors included information on the mechanisms of aging. In the following section, they described the possibility of using selected monoterpenes to delay the aging process by quoting the results of a number of recent studies focusing on both the compounds with the expected effects, as well as presenting their mechanism of action. Moreover, the chemical formulas of the compounds with anti-ageing properties as well as short information on isolation techniques from plants are also included. The article should be published because of the important aspects raised in it related, as mentioned by the authors, to the extension of life and the increasing number of elderly people suffering from various diseases related to, among other things, ageing.

However, the citation of literature in the text should be slightly corrected in accordance with editorial requirements (lines 173, 176, 179, 185, Table 4, 514, 563, 568, 575).

We truly appreciate the kind comment on our revision. We have carefully confirmed all the references along the manuscript and corrected those that did not match the editorial requirements. Accordingly, the list of references at the end of the manuscript was updated.

Reviewer 3 Report

Comments and Suggestions for Authors

The manuscript aims to describe the available data about monoterpenes and essential oils on ageing and related mechanisms. 

After a brief introduction and description of the topic (ageing and volatile monoterpenes), the authors describe available data based on hierarchical classification of ageing hallmarks, discussing results of preclinical data about isolated monoterpenes and essential oils on ageing-related molecular mechanisms and other effects not strictly related to ageing, such as in neoplastic models.

The manuscript is well-written and structured. The topic is extensively discussed.

I have some minor revisions to suggest to the authors:

I think a review of the title could improve its representativeness about the topic: what do you think about “Plant monoterpenes and essential oils as potential anti-ageing agents: insights from preclinical data”… or something similar?

A linguistic and punctuation revision could improve the manuscript. Some parts are difficult to read and some terms are widely repeated (such as namely or systematized).

I think this is not a systematic review of the literature on the topic as otherwise stated by the authors in the conclusions. However, given the good organization of the text, the manuscript could be implemented by a brief description of the method used to select sources of the literature and a flow chart numerically describing the articles obtained and discussed.

A brief hint about the bioavailability of these substances can be included in the discussion, maybe as an appraisal for future research. This aspect is crucial in shifting from preclinical to clinical trials. Also, the fundamental difference between treatment and prevention could be explored. A preventive treatment could be more effective than a treatment to solve an already unbalanced mechanism.

Please, use journal annotation for references at lines: 117, 173, 176, 179, 432, and Table 4

Please, specify “gut microbiota” because not always the heterogeneity of the microbe population is an advantage (as happens in the oral cavity)

Author Response

The manuscript aims to describe the available data about monoterpenes and essential oils on ageing and related mechanisms. After a brief introduction and description of the topic (ageing and volatile monoterpenes), the authors describe available data based on hierarchical classification of ageing hallmarks, discussing results of preclinical data about isolated monoterpenes and essential oils on ageing-related molecular mechanisms and other effects not strictly related to ageing, such as in neoplastic models. The manuscript is well-written and structured. The topic is extensively discussed.

I have some minor revisions to suggest to the authors:

I think a review of the title could improve its representativeness about the topic: what do you think about “Plant monoterpenes and essential oils as potential anti-ageing agents: insights from preclinical data”… or something similar?

We totally agree with the suggestion. We considered the proposed title as it better represents the content of the review.

A linguistic and punctuation revision could improve the manuscript. Some parts are difficult to read and some terms are widely repeated (such as namely or systematized).

We appreciate the comment. We have carefully reviewed the manuscript, correcting some linguistic aspects and included some alteration to avoid excessive word repetitions.

I think this is not a systematic review of the literature on the topic as otherwise stated by the authors in the conclusions. However, given the good organization of the text, the manuscript could be implemented by a brief description of the method used to select sources of the literature and a flow chart numerically describing the articles obtained and discussed.

We acknowledge the comment. Indeed, we agree that our review is not a conventional systematic compilation of available data. In order to provide the readers with a better understanding of how the studies were selected, we have included a more detailed explanation on this topic in section 3.2.

A brief hint about the bioavailability of these substances can be included in the discussion, maybe as an appraisal for future research. This aspect is crucial in shifting from preclinical to clinical trials. Also, the fundamental difference between treatment and prevention could be explored. A preventive treatment could be more effective than a treatment to solve an already unbalanced mechanism.

Thank you for the suggestions. We have included more information on these matters in the discussion section.

Please, use journal annotation for references at lines: 117, 173, 176, 179, 432, and Table 4.

We have carefully confirmed all the references along the manuscript and corrected those that did not match the editorial requirements. Accordingly, the list of references at the end of the manuscript was updated.

Please, specify “gut microbiota” because not always the heterogeneity of the microbe population is an advantage (as happens in the oral cavity).

We appreciate the suggestion. The word gut was included to avoid misunderstandings.